

# Spatio-temporal observations of nocturnal low-level jets and impacts on wind power production

Eduardo Weide Luiz[1,2] and Stephanie Fiedler[1,2]

[1]Institute for Geophysics and Meteorology, University of Cologne, Cologne, Germany
[2]Hans-Ertel-Centre for Weather Research, Climate Monitoring and Diagnostics, Cologne/Bonn, Germany

**Correspondence:** Eduardo Weide Luiz (eweidelu@uni-koeln.de)

**Abstract.** A challenge of an energy system, that nowadays more strongly depends on wind power generation, is the spatial and temporal variability of winds. Nocturnal low-level jets (NLLJ) are typical wind phenomena defined as a maximum in the vertical profile of the horizontal wind speed. A NLLJ has typical core heights of 50–500 m above ground level (a.g.l.), which is in the height range of most modern wind turbines. This study presents NLLJ analyses based on new observations from Doppler wind LiDARs. The aim is to characterize the temporal and spatial variability of NLLJs on the mesoscale and to quantify their impacts on wind power generation. The data was collected during the Field Experiment on Submesoscale Spatio-Temporal Variability (FESSTVaL) campaign from June to August 2020 in Lindenberg and Falkenberg (Germany), located at about 6 km from each other. Both sites have seen NLLJs in about 70% of the nights with half of them lasting for more than 3 hours. Events longer than 6 hours occurred more often simultaneously at both sites than shorter events, indicating the mesoscale character of very long NLLJs. Very short NLLJs of less than one hour occurred more often in Lindenberg than Falkenberg, indicating more local influences on the wind profile. We discussed different meteorological mechanisms for NLLJ formation and linked NLLJ occurrences to synoptic weather patterns. There were positive and negative impacts of NLLJs on wind power that we quantified based on the observational data. While NLLJs increased the mean power production by up to 80%, the stronger shear in the rotor layer during NLLJs had also negative impacts. The impacts of NLLJs on wind power production depended on the relative height between the wind turbine and the core of the NLLJ. For instance, the mean increase of the estimated power production during NLLJ events was about 30% higher for a turbine at 135 m a.g.l. compared to one at 94 m a.g.l.. Our results imply that long NLLJs have an overall stronger impact on the total power production, while short events are primarily relevant as driver for power ramps.

## 1 Introduction

Renewable energy (RE) sources play an important role for meeting targets to mitigate climate change and to improve the access to electricity (Sadorsky, 2021). The RE sector is growing and it is expected that its consumption will experience a 6.9% compound annual growth rate in the new policies scenario between 2014 and 2040 (OECD/IEA, 2017). On a global scale, wind power has the potential to cover more than one third of the global energy demand until 2050 (IRENA, 2019). In the European Union, 80% of the newly installed power capacities is RE technology and wind power may become a main energy





source after 2030 (OECD/IEA, 2017; Ziemann et al., 2020). In Germany, the share of RE to the overall power consumption is increasing continuously and largely stems from wind turbines already. A challenge for the success of the energy transition is the dependency of wind power production on meteorological conditions that vary in time and space (Frank et al., 2021; Druecke et al., 2021). This may cause power ramps associated with short-term increases or decreases in wind power production and variability on larger spatio-temporal scales associated with meso- to synoptic-scale weather phenomenon.

This article focuses on nocturnal low-level jets (NLLJs) and their impacts on wind power. A NLLJ is a maximum in the vertical profile of the horizontal wind speed in the lower troposphere with a typical core between 50-–500 m above ground level (a.g.l.) (Ziemann et al., 2020; Shapiro and Fedorovich, 2010). A past study of NLLJs in western Germany indicates that more than 16% of NLLJs have a core below 200 m a.g.l., that is in the height range of most wind turbines (Marke et al., 2018). Typical hub heights of onshore wind turbines are 80 m to 140 m a.g.l. with rotor diameters of 80–118 m a.g.l. (Rohrig

et al., 2019). A precise characterization, forecast and quantification of the uncertainty related to the wind speed at hub heights are crucial for wind power applications (Mirocha et al., 2016) and are needed for grid planning, financial calculations of operators (Rohrig et al., 2019) and site assessments for investors (Ziemann et al., 2020). In addition to the direct increase of the wind speed during NLLJ events, typically allowing increased power output (Abkar et al., 2016; Sharma et al., 2017), the NLLJ related vertical wind shear (speed changes with height) and veer (directional shift with height) have also impacts on

wind turbine power and reliability (Peña Diaz et al., 2012). The wind turbines may, for example, experience suboptimal or superoptimal power production, depending on shear and veer, getting to values up to 5% of the rated power for single 1.5 MW utility-scale turbines (Wharton and Lundquist, 2012; Vanderwende and Lundquist, 2012; Murphy et al., 2020). Moreover, they impose additional static and mechanical loads on the rotor blades and shift wind turbine vibrations to higher amplitudes (Gutierrez et al., 2016), counteracting the wake effects (Ziemann et al., 2020; Doosttalab et al., 2020). Negative shear (decrease

of the wind speed with height) produced by NLLJs at lower heights can also negatively affect wind turbines, which are usually designed assuming positive shear (Gutierrez et al., 2016). Partial presence of negative shear (and partially positive) in the rotor layer can slightly reduce the probability of damaging loads (Gutierrez et al., 2017), indicating that taller wind turbines, with rotors more often within the negative shear region of NLLJs, can be beneficial. The veer impacts are also directly related to the direction of the veer and the rotation of wind turbines (Englberger et al., 2020). Therefore, understanding positive and negative

NLLJ impacts on wind turbines with different configurations plays an important role for the wind power industry.

NLLJs are a common phenomenon. Some NLLJ climatologies indicate a frequency of occurrence of about 10 to 50% of the nights, depending on the location and the identification criteria (Baas et al., 2009; Lampert et al., 2016). NLLJs were detected at one site in northern Germany, from May 2001 to April 2003, in 19 out of 29 different European synoptic weather patterns (Emeis, 2014), using the classification "Grosswetterlagen" (James, 2007). Different meteorological conditions can generate

NLLJs. The classically described development mechanism is associated with the decoupling of nocturnal winds from the surface friction by the formation of a near-surface temperature inversion (Blackadar, 1957). This condition typically happens at night, particularly on days with little cloud cover that allows strong radiative cooling of the surface. The classical theory describes the NLLJ formation using the concept of an inertial oscillation, a process tied to the decoupling of the air flow from surface friction. The associated weaker dynamical friction due to reduced eddy viscosity enables an acceleration of the





air aloft (Ziemann et al., 2020; Fiedler et al., 2013), with the development of a pronounced super-geostrophic wind speed maximum in the course of the night (Shapiro and Fedorovich, 2010). Time and strength of the wind speed maximum depend of the geographical position and the time of decoupling. NLLJs also depend on the large-scale horizontal pressure gradient, commonly expressed as geostrophic wind. A near-surface temperature inversion paired with a sufficiently strong geostrophic wind can occur, for instance, at the edge of a mobile high pressure system with an approaching extra-tropical cyclone. NLLJs typically start developing around sunset, coinciding with the development of a near-surface temperature inversion, and decaying with the onset of vertical mixing during the morning of the following day (Blackadar, 1957; Sisterson and Frenzen, 1978; Van de Wiel et al., 2010; Beyrich, 1994). Also, due to its lower intensity in the upper part of the daytime boundary layer and the largest ageostrophic wind component near ground, NLLJs occur first at higher heights, descending with time while increasing their strength (Beyrich et al., 1997).

Other NLLJ driving mechanisms than inertial oscillations are known. For instance, a NLLJ profile can be detected when an aged cold pool, e.g., generated by downdraft from deep moist convection, glides up over a radiatively formed stable boundary layer. The cold moist air settles then above the nighttime temperature inversion, where a NLLJ forms as the result of reduced frictional deceleration (Heinold et al., 2013). Another possibility is when a near-surface temperature inversion is formed by warm air advection over relatively cooler near-surface air. This happens when air from land is advected over a relatively cold sea during late spring or summer, which is important to offshore wind turbines (Kalverla et al., 2019; Svensson et al., 2019).

There is no past observational study that explores the driving mechanisms of NLLJs with detailed analyses of their duration, meso-scale extent and impacts on wind power. Past works on NLLJ characterization often had the limitation of having rarely measurements of vertical profiles for wind speed and direction on mesoscales, i.e. a few kilometers with a temporal resolution of minutes to hours. Wind properties are routinely measured at single stations that are often located hundreds of kilometers apart, such that mesoscale characteristics in space can not be analyzed. Such measurements are usually also limited to the height of meteorological masts with heights typically up to 100 m and sometimes up to 300 m, that are insufficient to fully characterize NLLJs. Another limitation is the representation of NLLJs in atmospheric models, e.g., those used for numerical weather predictions (NWP) and reanalysis data. The models typically underestimate the strength and overestimate the height of NLLJs, while the wind veering between the surface and the top of the boundary layer is underestimated (Sandu et al., 2013; Svensson and Holtslag, 2009; Brown et al., 2005, 2008). Potential reasons are multiple and include the common artificial enhancement of the turbulent mixing during stable stratification to represent unresolved processes, e.g., vertical mixing associated with surface heterogeneity, gravity waves, and sub-grid scale variability (Sandu et al., 2013). Reanalysis data can share similar biases for the near-surface wind profile and coarse vertical resolutions are an additional contributing factor to those biases (Kalverla et al., 2019; Hallgren et al., 2020).

An opportunity to overcome these limitations is the deployment of Doppler wind LiDARs, which can continuously measure wind profiles with high vertical and temporal resolutions (Suomi et al., 2017). Doppler wind LiDARs emit laser beams in at least three different directions, which are scattered by aerosols. The measurements are used to determine the wind speed and direction based on the Doppler effect (Hallgren et al., 2020). Profiles of the mean wind speed are retrieved within the boundary layer and up to a few kilometres a.g.l., depending on the weather conditions. The quality of the wind retrieval depends on the



**Table 1.** Location, available number of days and periods with missing data of all three available LiDARs.

| LiDAR | Location | Total number of days | Missing Days |
|---|---|---|---|
| WL177 | Falkenberg | 84 | 24-31.08 |
| WL78 | Falkenberg | 69 | 18.06/11-31.08 |
| WL44 | Lindenberg | 91 | 24.08 |

amount of aerosols, affecting the strength of the back-scattered signal (Pearson et al., 2009). In this work, we made use of new data from Doppler LiDAR instruments at two sites in Eastern Germany. The two sites are about 6 km distant from each other allowing the study of mesoscale spatio-temporal characteristics of NLLJs from June to August 2020. We detected NLLJs with an automated algorithm in this dataset and systematically assessed their mesoscale characteristics, driving mechanisms, including synoptic weather patterns, and impacts on wind power production.

## 2 Data and Methods

### 2.1 FESST@home Data

The Field Experiment on Submesoscale Spario-Temporal Variability (FESSTVal, 2020) is a joint field campaign organized by the Hans-Ertel-Centre for Weather Research and involving different partners. The primary goal of FESSTVaL is measuring sub-mesoscale to mesoscale variability employing a measurement strategy to cover three main aspects: boundary layer patterns,
cold pools and wind gusts. Measurements for FESSTVaL were carried out between January 2019 and December 2021. It included a test campaign in 2019, a remote campaign FESST@home (due to the pandemic) at different locations in Germany during Summer 2020 and the main FESSTVaL campaign in spring and summer 2021. In the present study, we used the observational data from Doppler wind LiDARs deployed during FESST@home between June and August 2020. The LiDARs were installed and operated at the Lindenberg Meteorological Observatory – Richard Assmann Observatory (MOL-RAO) of the
German Weather Service (DWD) located in Lindenberg (Lat: 52.21° Lon: 14.13°) and Falkenberg (Lat: 52.16° Lon: 14.14°), which are located 6 km apart from each other in a rural area in Brandenburg, East of Berlin. For this work, data from three LiDARs were available: two (WL177 and WL78) located in Falkenberg and another (WL44) located in Lindenberg. All data have a temporal resolution of 10 minutes spanning partly different time periods (Table 1). The LiDARs operated in different measurement configurations. LiDAR WL44 used a method based on Päschke et al. (2015), WL177 operated in a gust mode
with focus on temporally highly resolved wind measurements, inspired by Suomi et al. (2017) and documented in Steinheuer et al. (2021a), and WL78 had a mode for measurements of turbulence parameters (Smalikho and Banakh, 2017). In addition to the LiDARs, Falkenberg also provides measurements from a meteorological mast up to 90 m a.g.l. for air temperature, wind speed and direction. The mast data was used for validating the LiDAR measurements and calculating the atmospheric stability.

All LiDAR data were put through quality controll. As first step, wind profiles with less than 90% of available data below
500 m were removed prior to the analysis. With this process, 3%, 4% and 6% of the profiles from WL44, WL177 and WL78,





were removed. In the second step, the LiDAR measurements were compared with the sonic anemometer data from 90 m a.g.l. in Falkenberg. Figure 1a shows the comparison for WL177. The other LiDAR in Falkenberg (WL78) had a similar behaviour (not shown). We can see a dependence of the data agreement on the wind direction. The behaviour is explained by the shadowing effect of the tower on the sonic anemometer for the 15º to 45º azimuth range. We therefore removed the data in this azimuth

range from the validation process. The resulting Pearson correlation coefficient R and the Mean Bias Error (MBE) between the anemometer and the LiDARs are shown in Table 2. Both WL177 and WL78 obtained measurements closely correlated with the anemometer measurements with R=0.93. The correlation for WL44 is expected to be lower, since the instrument is located in a distance of 6 km from the anemometer; even though R=0.85 is relatively large. So, considering the anemometer measurements as the ground truth, the second step of the validation consisted in excluding LiDAR profiles with a wind speed difference larger

than 20% from the anemometer measurements (as in Hallgren et al., 2020). About 4% of the remaining profiles from WL177 and WL78 were also excluded from this study. This step was not applied to WL44 in Lindenberg due to the large distance from the mast in Falkenberg.

**Table 2.** Pearson correlation coefficient (R) and Mean Bias Error (MBE) between the sonic anemometer and the different LiDARs at $\sim$90 m. On the left side, the measurements made in the azimuth range between $15°$ and $45°$ were excluded.

|  | 15 > Azimuth > 45 | | All Azimuths | |
|---|---|---|---|---|
|  | R | MBE (ms$^{-1}$) | R | MBE (ms$^{-1}$) |
| WL177 | 0.98 | 0.22 | 0.93 | 0.67 |
| WL78 | 0.99 | 0.10 | 0.93 | -0.07 |
| WL44 | 0.89 | -0.14 | 0.85 | -0.28 |

We further inter-compared the entire wind profiles from the LiDARs by calculating the Pearson correlation R between the LiDARs at different heights (Figure 1b). For this comparison, we interpolated all LiDAR data to the height levels of WL177,

with a vertical resolution of 26.5 m. The exact procedure is explained in more detail in the next section. Both LiDARs in Falkenberg (WL177 and WL78) have R > 0.98 between 53 m and 1000 m a.g.l., while R for both WL177 and WL78 against WL44 is lower, with R around 0.94 between 100 and 1000 m a.g.l.. Note that R near the surface (here $\sim$26 m) is lower in all comparisons, consistent with the stronger influence from surface differences. Due to the similar results and the larger amount of data from WL177, in comparison to WL78, we used WL177 in most analyses. The LiDAR in Lindenberg (WL44) was used

for assessing the spatial differences of NLLJs on the mesoscale. Hereafter, both WL177 and WL44 are called by the name of their location.

## 2.2 Automated NLLJ detection

We applied an automated detection algorithm for identifying NLLJs. Since there is no strict definition of a NLLJ, different methods for their identification have been proposed in the past. These include visual inspections of the profiles by eye (Emeis,

2014) and more objective methods using automated detection tools, e.g., a fall-off wind speed from the core of at least 2 ms$^{-1}$



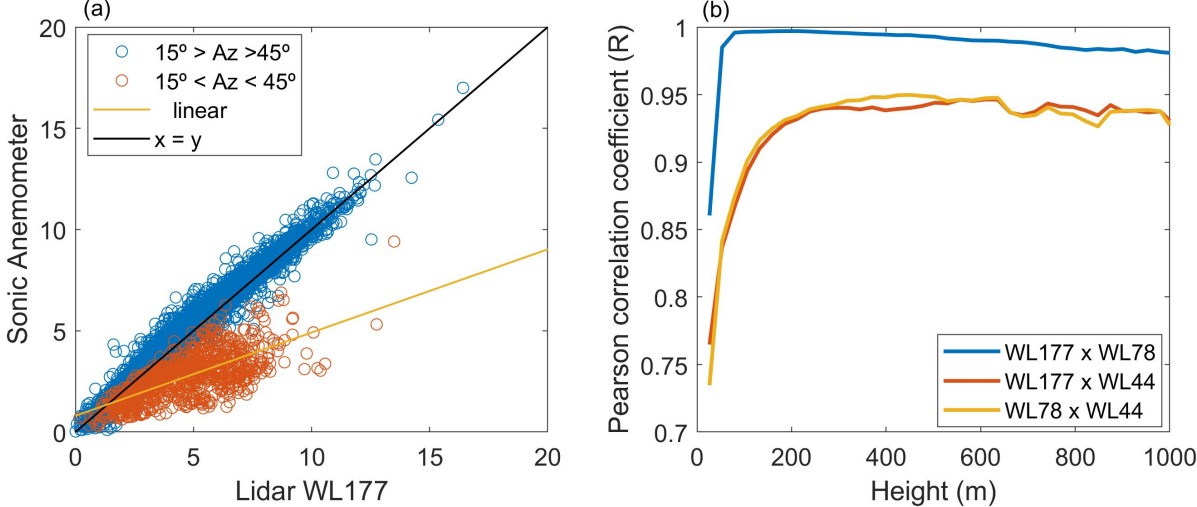

**Figure 1.** Data validation. Shown are (a) the horizontal wind speed at about 90 m a.g.l. from the the sonic anemometer against the LiDAR WL177, and (b) the Pearson correlation coefficient of the different LiDARs against the height a.g.l.. WL177 and WL78 are located in Falkenberg, while WL44 is operated ∼6 km away in Lindenberg.

compared with the minimum value in the wind profile above (Hallgren et al., 2020) and additionally below (Andreas et al., 2000; Banta et al., 2002). Others use relative values, like a 25% difference between the jet core and the minimum speed above (Baas et al., 2009; Wagner et al., 2019) or below (Tuononen et al., 2017). Some also include criteria for the maximum height and near-surface stratification (Fiedler et al., 2013), namely a jet core below 1500 m paired with a stably stratified surface layer 150 of at least 100 m depth and a vertical wind shear stronger than –0.005 s$^{-1}$ in the 500 m-deep layer above the core. The first criterion was used to reflect the reduced frictional effects in the nocturnal boundary layer as pre-requisite for the formation of NLLJs.

Here, for the definition of the NLLJ automated detection criteria, only valid profiles with solar height lower than 0° from Falkenberg were analysed. First, we smoothed the vertical profiles to avoid small and fast variability associated with turbulence 155 in the measurements. To that end, we interpolated the data vertically onto a new coarse-grained height profile as the average between every measurement height from WL177 in Fakenberg, giving us a vertical resolution of 26.5 m. Second, we calculated hourly moving averages for all wind profiles. Both these approaches successfully decreased the number of false detection of NLLJs, e.g., those that are extremely short or false alarms due to noise in the dataset.

Our automatic identification of NLLJs was based then on detecting the NLLJ core as the wind speed maximum in the lowest 160 500 m deep layer and a critical mean vertical shear in the wind speed in a 500 m deep layer above the NLLJ core. Following past studies, we required a minimum wind speed difference of 2 ms$^{-1}$ from the NLLJ core and the minimum above it and tested different mean threshold values for the vertical wind shear in the same layer. Figure 2 shows examples of NLLJ profiles identified with the tested shear criteria: < -0.0025 s$^{-1}$, < -0.005 s$^{-1}$ and < -0.0075 s$^{-1}$. The tests indicated a sensitivity of the number of detected NLLJs to the threshold for the shear criterion (Table 3), with the expected behaviour that a weaker threshold




resulted in more NLLJ identifications. Since the strongest threshold (-0.0075 s$^{-1}$) might have missed relevant NLLJs for energy

application, we chose the moderate setting of -0.005 s$^{-1}$ for the shear threshold in the NLLJ detection.

**Table 3.** Number of profiles flagged as NLLJ and the ratio of NLLJ profiles to all valid nocturnal profiles for different thresholds of the vertical shear in the horizontal wind speed.

| Shear threshold (s$^{-1}$) | Total number of NLLJ profiles | Ratio |
|---|---|---|
| -0.0025 | 1643 | 0.40 |
| -0.005 | 1410 | 0.35 |
| -0.0075 | 841 | 0.21 |

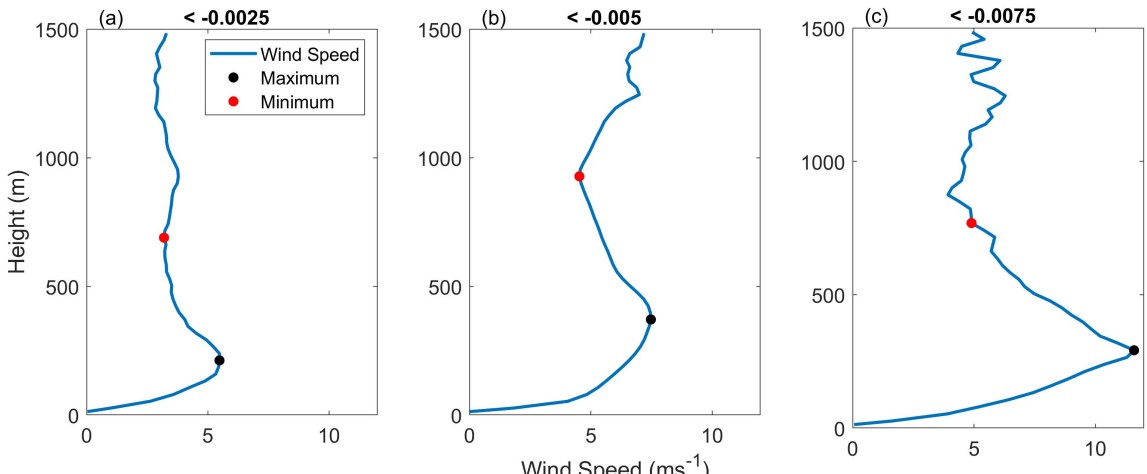

**Figure 2.** Examples of NLLJ profiles identified with wind shear between the jet core and the minimum above it with the thresholds < -0.0025 s$^{-1}$ (a), < -0.005 s$^{-1}$ (b), and < -0.0075 s$^{-1}$(c).

For an easier characterization of the NLLJs, we filled-up very short gaps in the NLLJ detection and removed very short NLLJ cases. First, we flagged all non-NLLJ profiles in between two NLLJ profiles also as a NLLJ, i.e., we filled-up 10 minutes gaps. This allowed us a better estimation of the duration of NLLJ events excluding very short perturbations. Second, we removed

NLLJ events of 20 minutes or less from our statistics, that were still not filtered out by the temporal smoothing process, to

focus events lasting longer than 20 minutes.

Up to here, all NLLJs were detected at nighttime, thus for solar heights below 0º. However, Figure 3 shows a large presence of NLLJs also during daytime. To account for the full lifecycle of NLLJs, we defined NLLJ nights as follows. Classical NLLJs can persist for a up to a few hours after sunrise. We therefore calculated the ratio between early morning and late afternoon

NLLJ detection for different solar heights, i.e., from 0º up to different positive solar heights. The chosen solar height for the

definition of the NLLJ night was 20º, with 92% of the NLLJ occurring during early morning. The detection of a small number





of NLLJs during late afternoon (8%) might be associated with cold pools. During the campaign, the mean morning time for 0º (20º) was 3 (5) UTC and for afternoon was 19 (17) UTC. All further analysis was made for solar heights smaller than 20º, which was equivalent to an average period of about 12 hours.

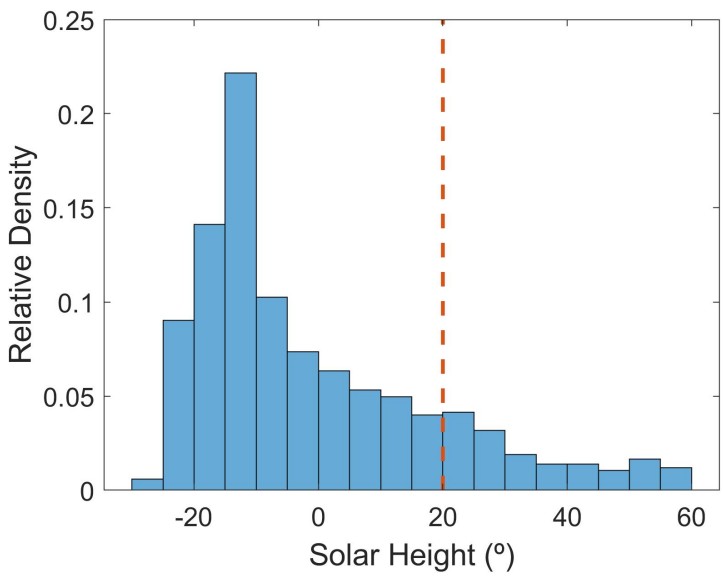

**Figure 3.** Histogram of the relative density of NLLJs by solar height. The red dashed line represents the selected maximum solar height for the definition of a "NLLJ night".

## 2.3 Atmospheric Stability

For the atmospheric stability and the turbulent mixing of momentum, two different metrics were calculated using data from the mast between 1 m and 10 m a.g.l. in Falkenberg. We calculated the temperature difference (dT) for the detection of near-surface temperature inversions as a measure of the static stability. Moreover, the Richardson number (Ri) was used to estimate the degree of turbulent mixing and the timing of downward mixing of momentum. Ri is calculated as the ratio of thermally induced turbulence and mechanically generated mixing by vertical wind shear:

$$Ri = \frac{g}{\theta} \frac{\partial\theta/\partial h}{(\partial V/\partial h)^2} \tag{1}$$

where $g = 9.81 ms^{-1}$, $\theta$ is the potential temperature, $V$ is the absolute horizontal wind speed and $h$ is the height a.g.l.. In this work, we limited RI values to $\pm 3.5$, since RI can have very large or very small values, depending on the atmospheric conditions.

Large Ri values imply that the stratification is stronger than shear-driven mixing. Unstable conditions and turbulent mixing are associated with negative Ri values. The critical threshold for the transition between stable and unstable conditions is about





**Table 4.** Characteristics of the Enercon E-126 and Vestas V112 Onshore wind turbines.

|  | Rated Power (kW) | Cut-in speed ($ms^{-1}$) | Cut-out speed ($ms^{-1}$) | Rated speed ($ms^{-1}$) | Diameter (m) | Hub height (m) |
|---|---|---|---|---|---|---|
| E-126 | 7580 | 3 | 34 | 16.5 | 127 | 135 |
| V112 | 3075 | 3 | 25 | 12 | 112 | 94 |

0.25 (Han et al., 2021), i.e., turbulence occurs when Ri is smaller than 0.25. Ri can be small below the core of a strong NLLJ despite the stable thermal stratification, indicating mechanically driven turbulence by the strong vertical wind shear (Gutierrez et al., 2016).

## 2.4 Wind power production

To quantify the impact of NLLJs on wind power production, we calculated the wind power production for two wind turbines: Enercon E-126 and Vestas V112 (Table 4). Their power curves (Figure 4) describe the wind power productions as function of wind speed. The cut-in and cut-out wind speeds mark their different wind speed ranges for power production. For the power simulations, the wind speed from each LiDAR profile was interpolated to the hub heights of the wind turbines. We further calculated the mean shear and veer in the mean rotor layer of ∼50-150 m, following Pichugina et al. (2017). The shear and veer was calculated as the mean wind difference per meter inside the rotor layer.

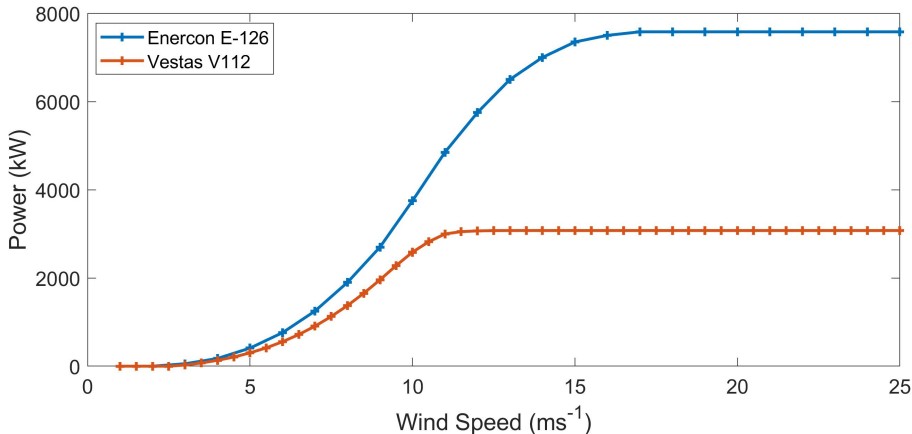

**Figure 4.** Power curves of the Enercon E-126 and Vestas V112 Onshore turbines.





# 3 Results

## 3.1 Statistics of NLLJs

The mean occurrence frequency of NLLJs were similar at both measurement sites, indicating NLLJs being a typical mesoscale

event. The frequency of occurrence of NLLJ profiles in Falkenberg was 29%, calculated by dividing all identified NLLJ profiles (1839) by the total of valid observed nocturnal wind profiles (solar height < 20º). The frequency of occurrence in Lindenberg was with 23%, slighly lower, with a total of 1607 profiles classified as a NLLJ.

Slightly stronger NLLJs occured in Lindenberg than Falkenberg, but the NLLJ core heights were very similar at both places (Figure 5) and indicated a clear increase of the wind speed with height (Figure 6). The mean wind speed at the jet core was of

8.5 (9.0) ms$^{-1}$, with a minimum of 2.5 (2.1) ms$^{-1}$ and a maximum of 15.2 (16.8) ms$^{-1}$ in Falkenberg (Lindenberg). The mean jet core height was 229.9 (224.1) m, with a minimum of 53 (79.5) m and a maximum of 477 (477) m. In most cases, the height of the NLLJ cores was below 300 m. Interestingly, the horizontal wind speed and height distributions of the NLLJs shown here were similar to statistics from Braunschweig (about 250 km to the West) during the summer months in 2013 (Ziemann et al., 2020). Even Pichugina et al. (2017) identified NLLJs with a mean wind speed of 9.4 ms$^{-1}$ and mean height 149.3 m a.g.l. at

the East coast of the USA.

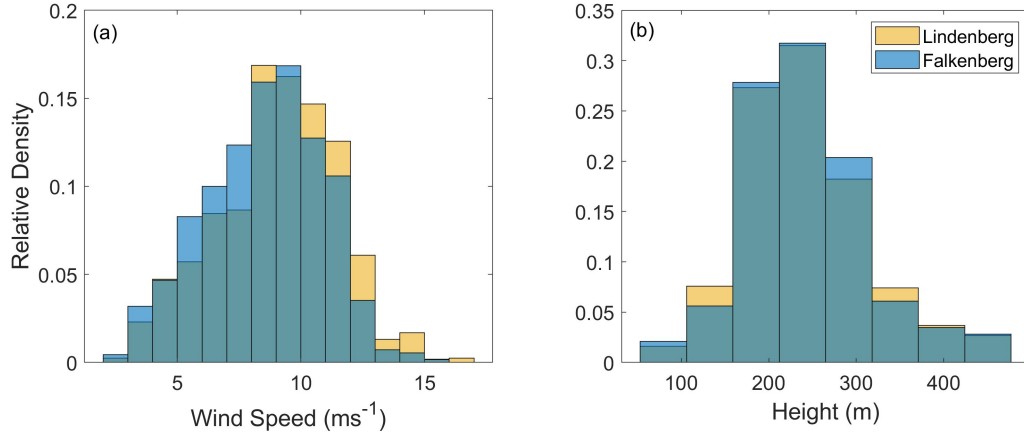

**Figure 5.** NLLJ core wind speed (a) and height (b) in Falkenberg (blue) and Lindenberg (yellow).

Looking at the duration of NLLJ events revealed more apparent differences across the two sites. Figure 7 shows a histogram with the length of the NLLJ events at both places. The minimum NLLJ duration was 30 minutes, due to the applied data filtering, and the maximum duration was about 11 hours, limited by the duration of the surface inversion. The time span with the largest number of NLLJ profiles was 0.5 to 3 hours. Lindenberg was affected by a larger number of NLLJ events shorter

than 1 hour.





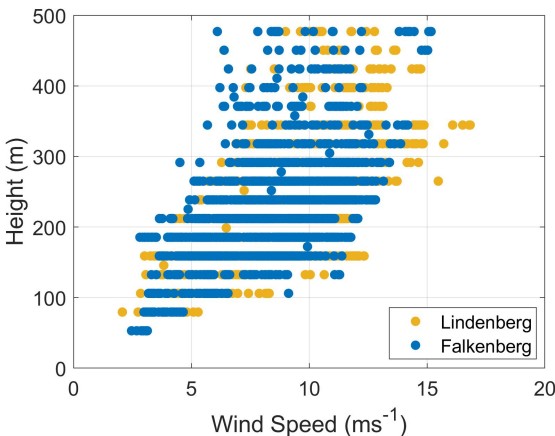

**Figure 6.** NLLJ core height against core wind speed color-coded for the two sites.

**Table 5.** Total number and percentage of nights with at least one NLLJ event, with temporal classification. The percentage for the four latter columns are calculated from the number of all NLLJ nights.

| Site | Number of nights with NLLJs | Very Short ($\leq$ 1 hour) | Short ($\leq$ 3 hours) | Long (> 3 hours) | Very Long (> 6 hours) |
|---|---|---|---|---|---|
| Falkenberg | 61 (73%) | 5 (8%) | 30 (49%) | 31 (51%) | 17 (29%) |
| Lindenberg | 58 (64%) | 11 (19%) | 30 (52%) | 28 (48%) | 14 (24%) |

### 3.1.1 Short vs. long events

Comparing the NLLJ wind speeds and heights for long events (> 3 hours) and short events ($\leq$ 3 hours) in both sites highlighted different behaviours. The mean jet core wind speed of long events was stronger 7.8 (8.4) ms$^{-1}$ against 6.1 (7.0) ms$^{-1}$ in Falkenberg (Lindenberg), but the extremes were more frequent for short events (Figure 8). The highest NLLJs were also the

ones that were typically short. Also, the mean core height with 222.9 (218.8) m for long NLLJs was a bit lower than for short events, with 244.3 (236.0) m in Falkenberg (Lindenberg). From Figure 8 we can also see that the longer events had narrower distributions.

Table 5 shows the number of nights with at least one NLLJ event and the duration of the longest event from each night. We can see that 73% (64%) of the nights presented at least one NLLJ event, in Falkenberg (Lindenberg), with a clear division of

about half of those nights having events longer than 3 hours. The presence of very long events (> 6 hours) was similar at both locations with 24% and 29% of the NLLJ nights, but the number of nights with only very short events ($\leq$ 1 hour) was more than doubled in Lindenberg (11) compared to Falkenberg (5).

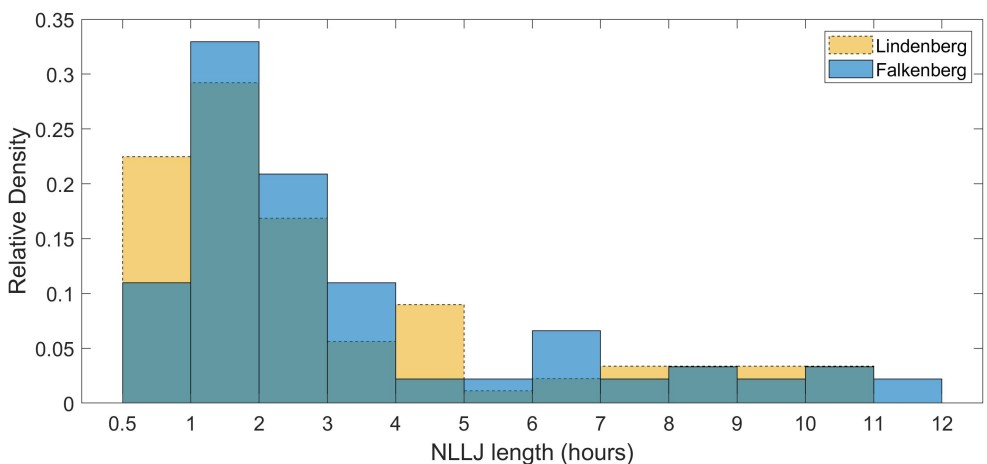

**Figure 7.** Lengths of NLLJs color-coded for the two sites. The data filtering implies that events shorter than 30 minutes are removed priot to the data analysis.

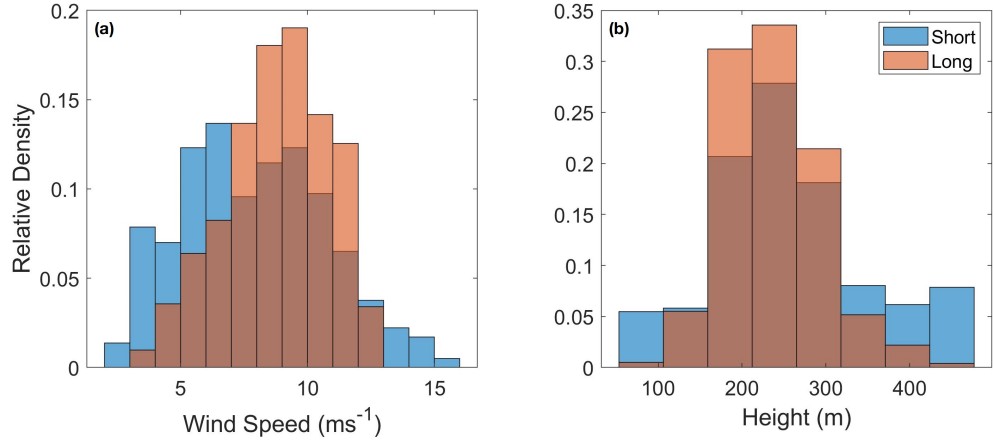

**Figure 8.** Frequency of NLLJ core wind speeds (a) and heights (b) in Falkenberg during short ($\leq$ 3 hours) and long events (> 3 hours).





### 3.1.2 Co-occurring events

Furthermore, we compared co-occurring NLLJs from both LiDARs. This assessment was inspired by the example of a long

NLLJ that occurred during the night 27–28 June 2020 at both locations, but developed differently over time (Figure 9). While the nocturnal development was very similar between 22 UTC and 02 UTC, there were differences during the early morning. Falkenberg saw a single long NLLJ that persisted until around 02 UTC, when the NLLJ detection in Lindenberg was interrupted. In Lindenberg, the NLLJ was detected again around 03 UTC, but not in Falkenberg. Such differences are associated with intermittent mixing and reflect the local differences in meteorological conditions. To better understand such effects, we

assessed the spatial differences of co-occurring NLLJ profiles at both sites next.

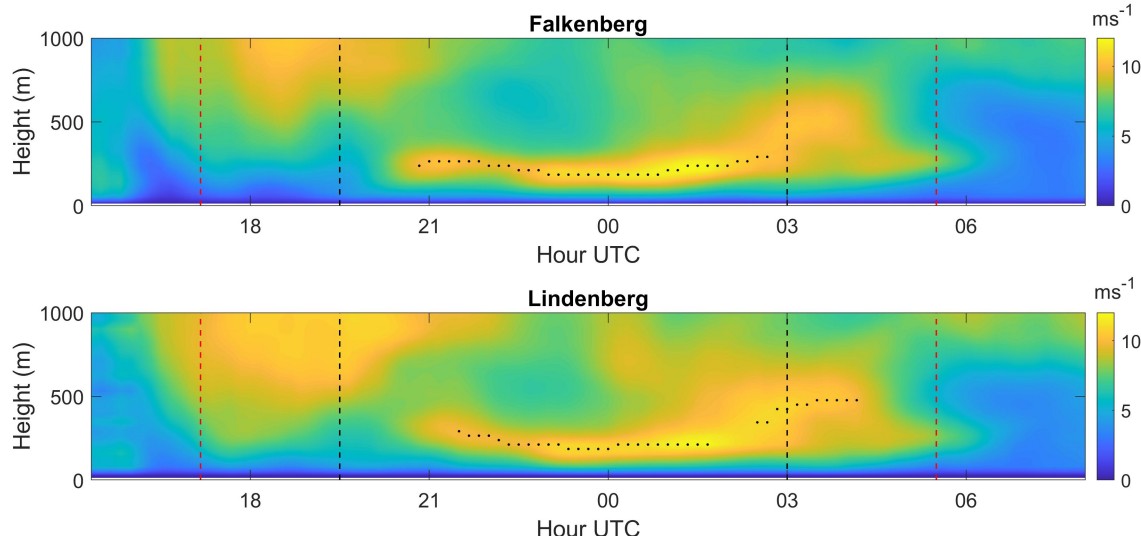

**Figure 9.** Absolute wind speed (m/s) during the night between 27–28 June 2020 in Falkenberg and Lindenberg as function of time and height a.g.l.. The black dots mark the NLLJ core and dashed black (red) lines the time of $0°$ ($20°$) solar elevation.

We performed a statistical comparison of the core wind speed and height for co-occurring NLLJ profiles at the two sites (Figure 10a–b). While the wind speeds were often similar at both places, the heights had larger differences between the two places. From the NLLJ profiles identified in Falkenberg, 75% were also identified in Lindenberg, while for the opposite it was 87%, underlining the often co-occurrence of NLLJ on the mesoscale. Restricting the analysis to short NLLJ events, the

percentages were smaller, with values of 48% and 53%. This is to be expected since short NLLJ events can be associated with intermittent developments of long NLLJs (as in Figure 9) or driven by density currents from convective cold pools that may not affect both sites simultaneously. This behaviour was also reflected in the lifetime of co-occurring NLLJs at the two sites (Figure 10c). Co-occurring NLLJs lasting for more than six hours had typically similar lifetimes at both places. This is consistent with the mesoscale spatial extent of NLLJs being driven by reduced frictional coupling of the wind field with surface friction in an

inertial oscillation.





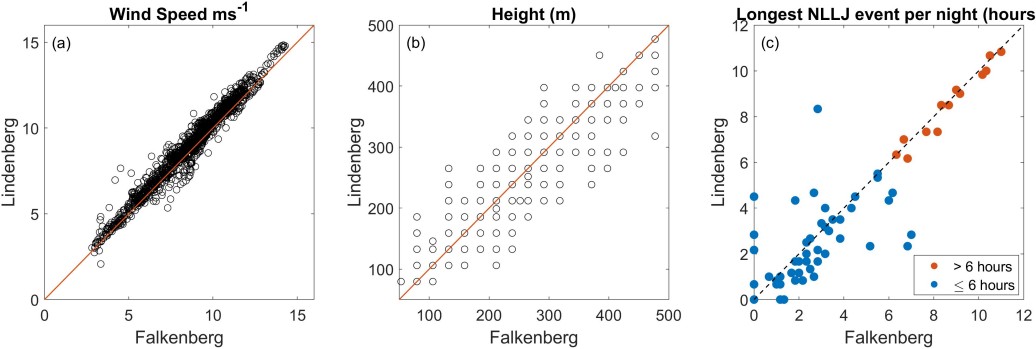

**Figure 10.** Characteristics of co-occurring NLLJs at the two sites. Shown are (a) core wind speed, (b) core height and (c) lifetime of the longest NLLJs per night. In (c), events longer than 6 hours in both sites are shown in red.

### 3.1.3 Temporal development

Looking at long NLLJ events in more detail revealed a nocturnal development similar to what one expects from inertial oscillations. Figure 11 shows the diurnal cycles of long NLLJs, including distributions of the jet core speed and height as well as the normalized mean Ri (nRi) and dT (ndT) between 1 and 10 m a.g.l.. The normalization was made by dividing the mean

hourly values by their maximum positive values in order to better illustrate the diurnal cycle. The development of long NLLJs depended on the presence of the near-surface temperature inversion. Take for instance the temperature gradient ndT in Figure 11a. After sunset, ndT increased indicating the development of a surface-inversion, leading to a reduction of frictional effects on the air flow in some distance from the surface. Stable stratification prevailed until the early morning hours, when NLLJs also frequently occurred. We also saw an increase of the mean core wind speed over time up to about 01 UTC, when it started

to decrease until the early morning. Up to 23 UTC, the mean jet height decreased and remained about constant thereafter. This behaviour is broadly consistent with the process expected for an inertial oscillation, but there are also differences as to be expected from the idealized assumption of stationarity in the theory. For example, longer periods of increasing wind speed is predicted from the theory, coinciding with half a period of the oscillation in mid-latitudes (approx. 8 hours (Van de Wiel et al., 2010)). This difference from the theoretical duration was apparent, especially for the short events. In addition to non-

stationarity, reasons for differences between the measurements and the theory are the neglection or simplification of the smaller, yet still present, frictional effects (Blackadar, 1957; Van de Wiel et al., 2010). In reality, NLLJs are affected by intermittent mixing events, i.e., temporally variable frictional coupling of the jet with the surface layer and changes of the geostrophic wind along with the synoptic-scale weather conditions.

    The wind directions changed in the course of the development of NLLJs, but not as systematic as one would expect for a

classical inertial oscillation. We selected the data from Lindenberg for quantifying the changes in wind direction in the core of the NLLJs. Most NLLJ events did not show large changes in wind direction in the course of their lifetime, falling between -40° to 40° total directional changes calculated as the total wind direction shift in the NLLJ core. NLLJs more frequently had a clockwise change (positive values in Figure 12a) than anticlockwise. An inertial oscillation causes the wind field to veer in the

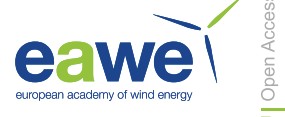


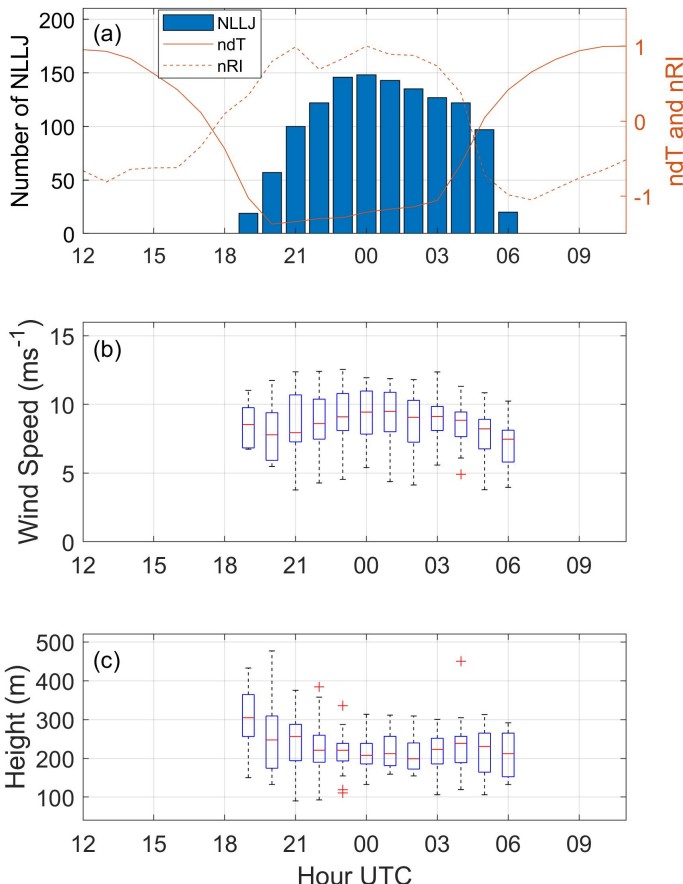

**Figure 11.** Diurnal cycle of NLLJs in Falkenberg. Shown are the hourly (a) number of long NLLJs, (b) their core wind speed and (c) their core height. Lines in (a) show the hourly averaged and normalized Ri (nRi) and dT (ndT).

course of the night. One therefore expects a larger directional shift of the NLLJ wind, the longer the events are. We identified

a total shift in the direction of mean relative (absolute) values of 7° (34°) for short events, 19° (107°) for long events and 36° (132°) for very long events, consistent with veering in an inertial oscillation. Indeed, some of the longest events (marked by dashed lines in Figure 12a) had large wind directional shifts, consistent with the theory of an undisturbed inertial oscillation.

The turning of the wind is very clear for the longest NLLJ event in Lindenberg during the night of 12–13 August 2020 (Figure 12b). The NLLJ development began around sunset shortly after 18 UTC and ended due to turbulent mixing in the first

morning hours (04–06 UTC), indicated by the morning time increase in dT. The perturbations in the turning of the wind are clearly related to fast changes in the geostrophic wind at about 00 UTC. This relationship between wind turning perturbations and geostrophic wind fast changes also happens to other events from the campaign (not shown). This NLLJ event also led to supergeostrophic wind speeds consistent with the theory (Figure 12c). The supergeostrophy might at least in parts be explained by the slackening geostrophic winds in the course of the night. Thus, changes in the synoptic-scale conditions affected the





NLLJ development, although this event is close to characteristics expected from the theory. Other NLLJ events showed larger deviations from the theory (not shown), indicative of intermittent turbulence perturbing the nocturnal acceleration and turning of the wind field paired with temporal changes in the horizontal pressure gradients due to the evolving weather. The NLLJ of 12-13 August 2020 was associated with an anticyclonic south-easterly wind over the measurements sites due to a high pressure system to the Northeast.

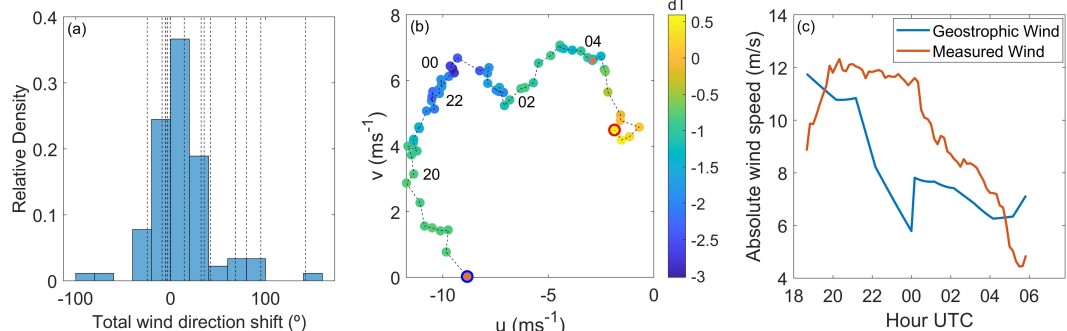

**Figure 12.** Directional shifts during NLLJ development. Shown are (a) Frequency of the total wind direction shift in the NLLJ cores for Lindenberg, (b) the zonal (u) and meridional (v) wind components during the NLLJ in the night of 12–13 August 2020 in Falkenberg, and (c) the actual and geostrophic wind speed. In (a) positive values mean a clockwise shift and dashed lines mark the total directional shift during the very long NLLJs. In (b) the color illustrates the vertical temperature change (dT) in the first 10 meters a.g.l., the blue (red) circle marks the start (end) of the NLLJ. The additional orange circles mark the sunset and sunrise, and the numbers are the UTC times. The geostrophic wind in (c) was calculated from ERA5 data (Hersbach et al., 2020) and interpolated to the measurement times.

### 3.1.4 Weather patterns

We systematically investigated the occurrence of NLLJs under different synoptic weather patterns. Using the "Großwetterlagen" of the German Weather Service (James, 2007), we detected NLLJs in 13 out of 16 different weather patterns occurrences (Figure 13). Three weather patterns had particularly favourable conditions for NLLJ development. The pattern with the largest amount of NLLJ events was the *Scandinavian High, Trough Central Europe* (HFz) pattern (Figure 14a). The most efficient pattern was the *Anticyclonic South-Easterly* (SEa, Figure 14b), with the highest overall probability, since all nine cases were associated with a NLLJ. Both weather patterns point to inertial oscillations being driven by the sufficiently large geostrophic wind at the margin of a high pressure system, paired with relatively dry conditions allowing stronger nocturnal radiative cooling of the surface. The two synoptic weather patterns are shown as composites in Figure 14a–b, indicating high pressure systems in Northern Europe and lower pressure towards the South resulting in large horizontal pressure gradient over the measurement site. The pattern with the largest number of short NLLJ events was the *Cyclonic Westerly* (Wz) (Figure 14c), pointing to NLLJs associated with the passage of low pressure systems. Northeast Germany was also found in a zone with strong pressure gradient during this weather pattern.

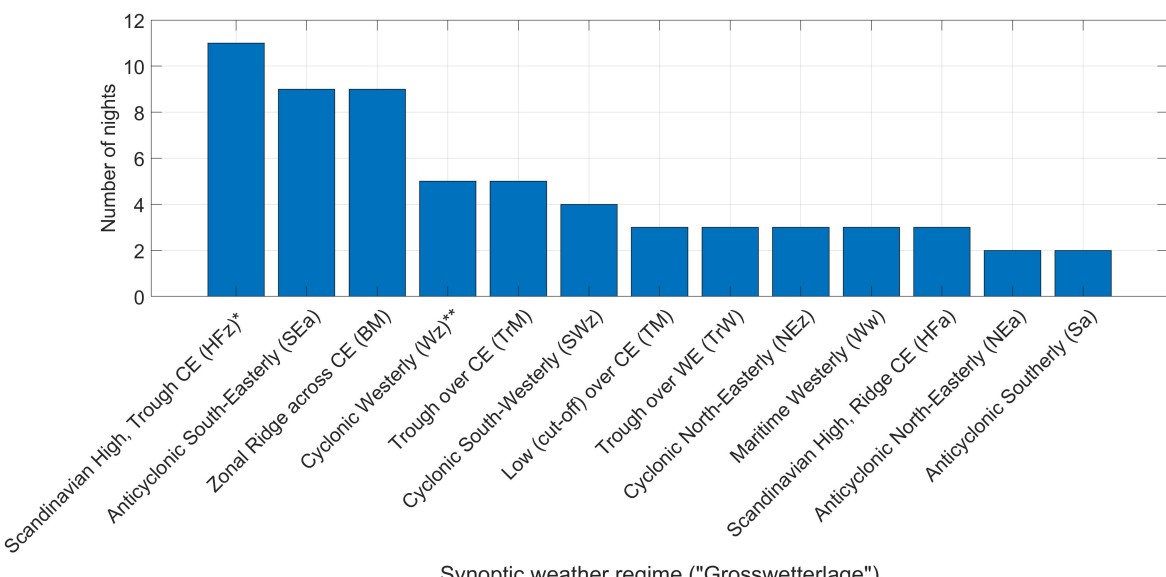

**Figure 13.** Number of Nights with NLLJs over Falkenberg as function of the European synoptic weather patterns (Großwetterlagen, see James (2007)). Legend: CE = Central Europe. WE = Western Europe. *Largest amount of long events. **Largest amount of short events.

These results are broadly consistent with findings by Emeis (2014), based on data from May 2001 to April 2003 in Hannover (about 300 km to the West from our sites). They indicate that a short period can be sufficient to assess NLLJs associated with
different weather patterns, although longer datasets would be needed for a full climatological assessment. It is interesting that we detected NLLJs during two patterns that were not detected in the previous study, namely *Cyclonic North-Easterly* (NEz) and *Anticyclonic North-Easterly* (NEa). These differences may be related to the different NLLJ identification methods, different locations or different time periods. In Emeis (2014), SEa also had a very high efficiency, however the *High Scandinavia-Iceland, Ridge Central. Europe* (HFNa), with the highest efficiency in his work, did not occur during our campaign.

## 3.2 Effects of NLLJ on wind power generation

### 3.2.1 Power generation

NLLJs have a strong effect on wind power generation since they increase the wind speed at rotor heights. We quantified these effects using wind data at 100 m a.g.l. from the LiDAR WL177 in Falkenberg, representative of typical rotor hub heights (Figure 15a). The frequency distribution of these hub wind speeds during NLLJ events is clearly shifted to higher values compared to
all data. For instance, the mean hub wind speed of $4.8 \, \text{ms}^{-1}$ from all data is smaller than the mean of $6.0 \, \text{ms}^{-1}$ during NLLJ events.

We calculated the wind power generation from two wind turbine models: Enercon E-126 and Vestas V112. The former has a much larger rated power and steeper increase in power generation with wind speed. We therefore normalized the wind



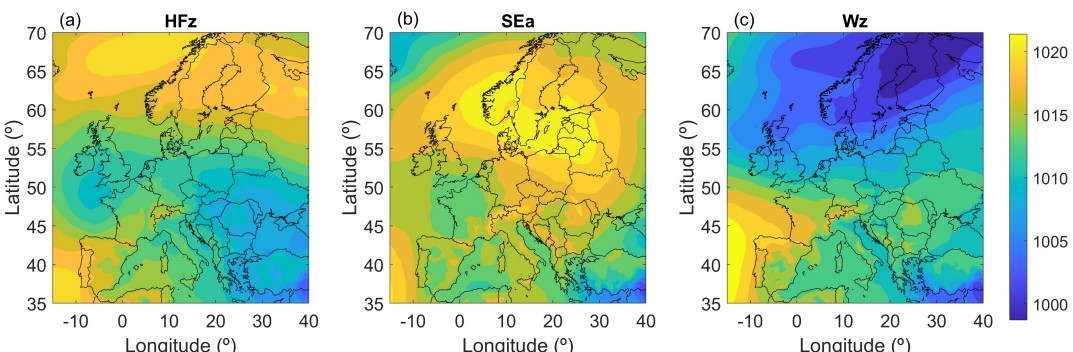

**Figure 14.** Weather patterns favourable for NLLJ development. Shown are composites of the 00 UTC mean sea-level pressure patterns during FESST@home for the weather patterns: (a) *Scandinavian High, Trough Central Europe* (HFz), (b) *Anticyclonic South-Easterly* (SEa) and (c) *Cyclonic Westerly* (Wz). The mean sea level pressure data was obtained from ERA5 reanalysis (Hersbach et al., 2020)

.

power estimates by the respective rated power of the turbine for a better comparability of the results (Figure 15b–c). The
results highlighted the shift of the frequency distribution of wind power generation to higher values during NLLJ events. The
distributions reflect the production in the lower half of the power curve since the wind speed rarely reached values needed for
yielding the rated power.

The additional power generation during NLLJs non-linearly depends on the turbine type. NLLJs increased the power gener-
ation of V112 (E-126) by 53% (80%) compared to all conditions. This implies absolute power values of 382.2 (583.3 kW/h)
for V112 and 788.2 (1412.2 kW/h) for E-126 during all conditions (NLLJs). In comparison, the changes in wind speed at rotor
heights were not as strong: 23% for V112 and 32% for E-126. This behaviour reflects non-linear dependencies of the power
generation on NLLJ occurrence and characteristics, depending on the turbine type and hub height. Power production from the
higher wind turbines with larger rated power therefore benefits more strongly from the occurrence of NLLJs.

Weak NLLJs (e.g. Figure 16d) were rare such that most NLLJs were strong enough to reach the cut-in wind speed and
allowed power generation . Our data presented such weak NLLJs in 12 individual profiles translating to 0.2% of all valid NLLJ
profiles for E-126 and 45 profiles or 0.7% for V112. These differences between the two turbine types are primarily explained
by the different rotor heights, since the cut-in wind speeds are identical.

### 3.2.2  Wind shear and veer through the rotor layer

While NLLJs are beneficial for the nocturnal power generation, their occurrence has also adverse implications for the turbines.
Wind shear and veer in the rotor layer, here defined as 50–150 m a.g.l., were considerable during the occurrence of NLLJs,
as indicated in the examples of wind profiles with different wind shear values (Figure 16). These are a case with (a) strong
positive shear without the presence of a NLLJ, (b) strong positive shear due to a NLLJ, (c) a NLLJ with the core inside the
rotor layer, causing positive shear below and negative shear above the core, and (d) a NLLJ with a core immediately below the





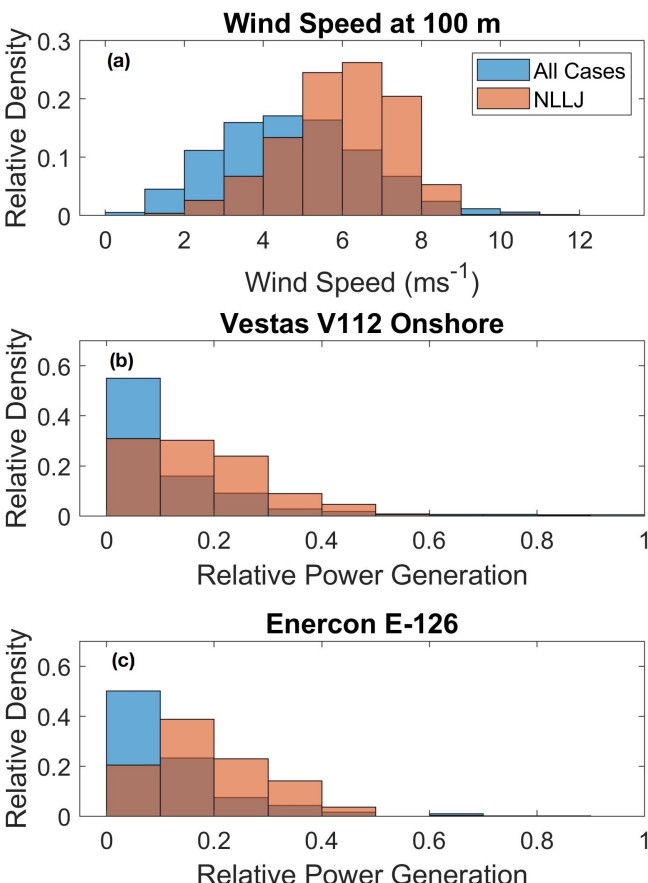

**Figure 15.** Wind power production associated with NLLJs. Shown are the frequency of (a) the wind speed at 100 m a.g.l., and the relative power production from (b) E-126 and (c) V112 for all cases (blue) and NLLJ conditions (red). The relative power generation was calculated relative to the rated power of the turbine types.





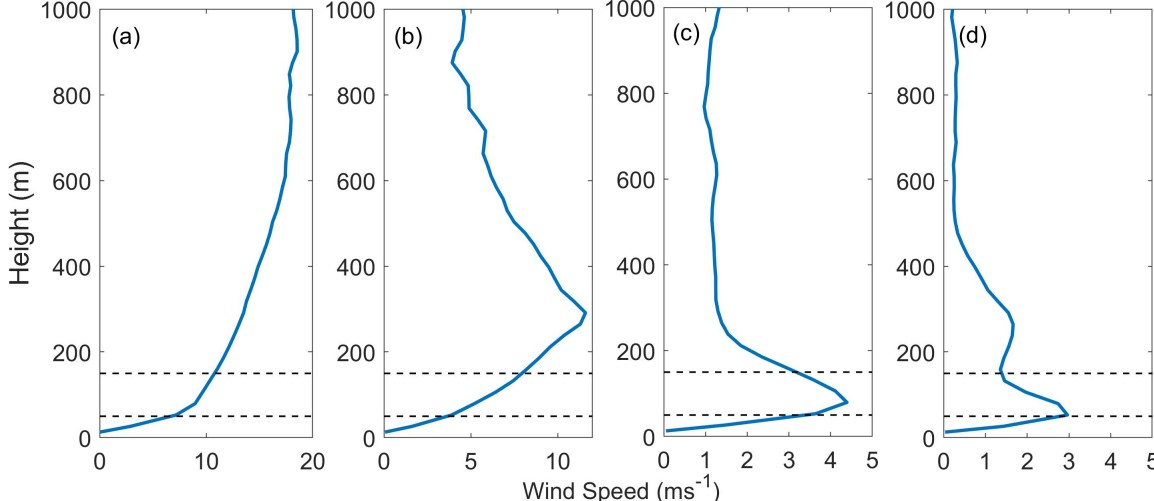

**Figure 16.** Examples of wind profiles with different vertical shears in the horizontal wind speed: (a) strong shear without the presence of a NLLJ, (b) strong shear due to a NLLJ, (c) a NLLJ with the core inside the rotor layer and (d) a NLLJ immediately bellow the rotor layer. The dashed lines mark the mean rotor layer of 50–150 m a.g.l..

rotor layer, which leads to negative vertical shear in the entire rotor layer. It is important to mention that when the NLLJ core
falls within the rotor layer care should be taken of how the shear and thus, the mechanical loading, is calculated. If we were to
calculate the mean shear taking positive and negative shear together, it would result in a false impression of small shear. We
therefore calculated the shear based on absolute values across the rotor layer for these conditions.

The shear was substantially larger during NLLJs events. The mean wind shear in the mean rotor layer had an increase of
about 67% during NLLJ events. A similar value was found for V112 since that turbine typce has a similar rotor layer. The
higher E-126 had a lower mean shear of 53%, but the increase during NLLJs was 84%. We rarely saw negative shear values
from cases like illustrated in Figure 16d and those that occurred were often below the cut-in wind speed, consistent with having
relatively few and weak NLLJs at very low levels.

Our results indicate that almost half (48%) of all cases with extreme wind shear coincide with NLLJs (Figure 17a). It
implies that NLLJs are as important as storms for causing strong wind shear. Out of all occurring NLLJs, 37% of NLLJs lead
to extreme shear through the rotor layer. We here defined extreme wind shear through the rotor layer as the 90% percentile,
following Debnath et al. (2021), which corresponded to a similar threshold of $0.0354\,\mathrm{s^{-1}}$. The extreme shear was primarily
explained by larger core wind speeds for NLLJs at similar heights, rather than stronger NLLJs at higher altitudes. This can be
seen by the similar distributions of the NLLJ height paired with a shift in the distribution of core wind speeds during NLLJs
with extreme shear (Figure 17b–c).

In addition to shear, the wind veer with height across the rotor layer has also implications for the mechanical loading on the
rotors. We therefore quantified the absolute wind veer across the rotor layer during NLLJ events (Figure 18). The magnitude
of the wind veer was typically larger during NLLJ events with a mean of 0.174°/m against 0.145°/m during all cases. It

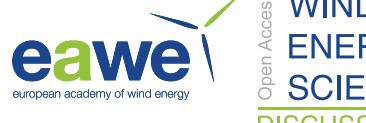

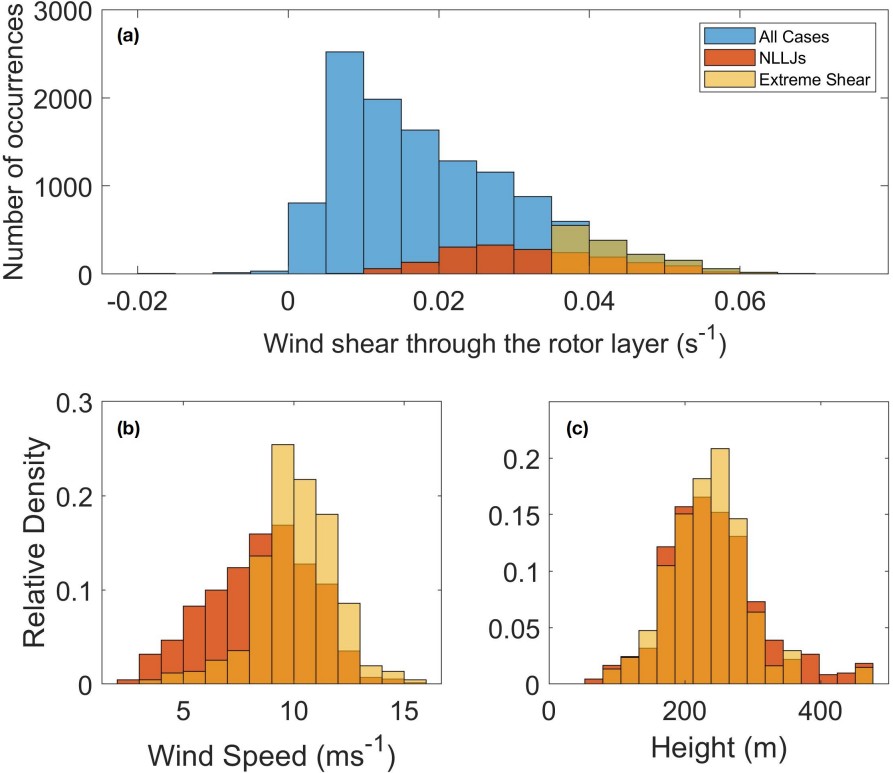

**Figure 17.** Vertical shear of wind speed in the rotor layer. Shown are (a) the number of occurrences of vertical shear in the rotor layer, and the frequency of (b) wind speed and (c) height. All conditions are shown in blue, NLLJs in red and extreme shear cases in yellow. Extreme shear was defined by the 90% percentile ($0.0354$ s$^{-1}$)

.

corresponded to a mean increase of the veer by about 20%. The results for V112 were similar and slight differences were seen for E-126. For E-126, the average veer was 11% lower than for V112 and the mean increase was 12% during NLLJs.

# 4  Discussion


NLLJs have positive and negative impacts on wind turbines. The wind power production clearly benefits from NLLJ occurrences and this positive impact increases with the height of the turbine. At the same time, the wind shear and veer increases with the height a.g.l.. These results indicate that installing higher turbines, that are able to benefit from this higher wind speed and are sufficiently stable to sustain high shear and veer, would be ideal for increasing the wind power production during stable


stratification of the near-surface boundary layer. Forecasts and reanalysis data do not fully characterize NLLJs with a sufficient accuracy. Monitoring vertical profiles of winds and temperature for site assessments is therefore important for planning new wind power installations.





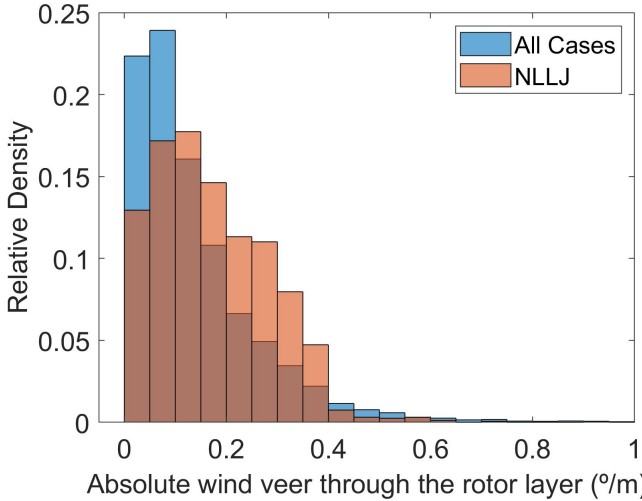

**Figure 18.** Wind veer in the rotor layer. Shown is the frequency of the absolute wind veer in the mean rotor layer (50–150 m a.g.l) during all nights (blue) and during NLLJs (red).

In this context, accurate representations of NLLJs in atmospheric models is of large importance for grid planning and power forecasting. Synoptic-scale weather patterns are no clear indication for NLLJ occurrence since in our results most patterns

could be associated with a NLLJ. Statistically, however, anticyclonic weather patterns stood out to favour NLLJ developments. These included weather patterns that can be connected to atmospheric blocking events. Atmospheric models are known to have difficulties to represent the correct onset and decay of such events (Tibaldi and Molteni, 1990; Lupo, 2021), which needs to be resolved in the future for better wind power forecasts. Studies on NLLJs with high temporal and spatial resolutions are useful to the extent that they help to better understand NLLJs and their driving mechanisms, evolution and connection with other

influences. To that end, the exploitation of the new FESSTVaL data from 2021 will be helpful to improve forecasts of NLLJs and associated wind power production.

## 5 Conclusions

This paper analyzes the spatio-temporal characteristics of Nocturnal Low Level Jets (NLLJ) from June to August 2020, during the FESST@home campaign, at two sites in Eastern Germany: Falkenberg and Lindenberg. In addition, the impact of NLLJs

on wind power production is quantified. NLLJs occurred in 64% and 73% of the nights, depending on the site. About half of them had lifetimes exceeding three hours.

Detailed analyses of the NLLJs at the two sites suggested different driving mechanisms for their development. Very long NLLJs occurred more often simultaneously at both sites, indicating their mesoscale character. Our results indicate that NLLJs with long lifetimes are driven by inertial oscillations perturbed by nocturnal changes in the synoptic-scale, horizontal pressure

gradient and intermittent turbulent mixing. This was further supported by the prevailing synoptic weather patterns for long



NLLJs. Many long NLLJs occurred during anticyclonic weather patterns as one would expect to be favourable for inertial oscillations. Shorter NLLJ events are more strongly affected by local conditions. For instance, short NLLJs occurred more frequently in Lindenberg, suggesting stronger local influences on the NLLJ development than in Falkenberg. Short NLLJs can be caused by an earlier breakdown of a NLLJ development, or are driven by other meteorological processes, such as jet profiles created by frontal passages or density currents from convective downdrafts. This was again consistent with the prevailing cyclonic weather synoptic-scale weather pattern for short NLLJs. However, it is clear from our analysis that the synoptic-weather pattern alone is no clear indicator for whether a NLLJ forms or not.

NLLJs can have both beneficial and adverse impacts on wind power production. NLLJs increased the nocturnal power generation, but also increased wind speed and directional shear across the rotor layer. The magnitude of the effects depended on the hub height of the turbine. We estimated a wind power production increase by 50% (80%) for hub heights of 94 (135) m a.g.l. during NLLJs. Out of all NLLJs, 37% lead to extreme wind speed shear across the rotor layer. Compared to all extreme shear cases, 48% were caused by NLLJs. It highlights the strong impact of NLLJs for generating low-level shear during summer in Germany. These results imply that power production with higher turbines would more strongly benefit from NLLJs, particularly when adverse effect of wind shear and veer from the NLLJs is considered in the turbine construction.

Taken together, long NLLJs driven by perturbed inertial oscillations have a larger importance for wind power production. Long events not only sustained high wind speeds in the rotor layer over a longer time period, but also had a mesoscale spatial extension, holding the potential to affect one or several wind parks at the same time. Differently, short NLLJs were often more local and do not as often affected different sites simultaneously. Short NLLJs are therefore expected to cause power ramps, i.e., fast increases and decreases in power production, increasing the spatio-temporal variability in power production. Future work will address the meteorological processes of NLLJs with longer datasets and for larger regions including offshore regions and complex terrain.

*Data availability.* The Doppler LiDAR data is available from Steinheuer et al. (2021b).

*Author contributions.* E.W. Luiz: Conceptualization, Writing - original draft, Validation, Methodology, Software; S. Fiedler: Conceptualization, Supervision, Review & editing, Methodology

*Competing interests.* No competing interests are present.

*Acknowledgements.* This work was carried out in the framework of the Hans-Ertel-Centre for Weather Research. We acknowledge the funding for the HErZ research area "Climate Monitoring and Diagnostic" (ID: BMVI/DWD 4818DWDP5A). We thank the instrument





operators for carrying out the measurements during FESST@home, and the German Weather Service and Julian Steinheuer for providing the Doppler LiDAR wind data.



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
