# Peer review of "Spatio-temporal observations of nocturnal low-level jets and impacts on wind power production"

_Wind Energy Science, 2022_

## Author Response (AR1)

**Point-by-point reply to comments on manuscript with the title "*Spatio-temporal observations of nocturnal low-level jets and impacts on wind power production*"**

We thank the reviewers for the appraisal of our manuscript. The comments and suggestions were helpful to further improve the article. Please find below our replies (in blue) to the reviewer's comments (in black).

Reviewer 1:

Specific comments:

Some minor issues that could be improved:

1. Adequacy of title and scope: The scope of the paper is mostly focused on NLLJ characterizations and impact on wind profiles (shear and veer). Wind power production is assessed for NLLJ / Non-NLLJ periods, but this information is part of local wind climatology and the results were somehow expected. I suggest further examination on the unfolding questions that arise: How the shear and veer affect the wind power curve? What fraction of the wind power potential can be attributed to NLLJs? How it could support variability studies in longer time scales (even climate change)? How the weather patterns could help?

AC: We address these questions in the manuscript as follows:

- The analysis of the effect of shear and veer has been explored in the literature. We now added more discussion of this aspect in the manuscript (lines 378-384): "In addition, due to the changes in the expected power output (Murphy et al., 2020), it is clear that ignoring the shear in the rotor layer and only measuring wind at the hub height would affect wind power estimates. Another important effect is that the NLLJ wind profiles are very different from the often assumed logarithmic or power-law profiles used to extrapolate winds from 10 m to hub heights (Gualtieri, 2019; Hallgren et al., 2020). These approximations can lead to large errors in estimates of wind power potential, especially during NLLJs when the vertical wind profile strongly differs from average conditions. This is a particular problem for wind power estimates in locations with frequent occurrences of NLLJs."

- In the results section, we now also comment on the fraction of the NLLJ profiles to the total power production potential during the campaign (lines 332-335): "From the estimated total wind power potential during the campaign, 24% (28%) was generated during NLLJs conditions for the V112 (E-126)".

- The point of analyzing longer time scales is of interest and subject of ongoing work. The weather patterns used here are at least for the campaign of limited use for predicting NLLJ formation. This is in parts due to the influence of the

boundary layer development. As such it might be interesting to explore different techniques, like machine learning, to identify connections of NLLJs and their preconditions in the future as aid for their better prediction. The following was included in the discussion section (lines 392-397):

"Studies about NLLJs with high temporal and spatial resolution are useful to the extent that they help to better understand their driving mechanisms including the evolution of boundary layer characteristics and the influence of meso- to synoptic-scale weather developments. The present study is one step in that direction, but more can be done to support the energy transition using more wind power. Other techniques, including machine learning, may be an important tool to understand their large-scale driving mechanisms. One such aspect is to what extent NLLJs change with global warming that is currently not understood, but important for the operation of a future energy system."

2. Assumption of a mesoscale process: The signals from NLLJ in the nearby sites are similar what makes the assumption of a mesoscale process robust (and I tend to agree) but I'm not sure if this (6 km apart observation) is enough evidence for this statement. Surely there is a broad literature supporting these findings, and I suggest to revise the text (mostly discussion and conclusions) and rely more in this literature and not only on the concurrent observations.

AC: Thanks, we now revised the text to explicitly state the range of the mesoscale to be more specific. Our measurements give evidence for meso-gamma scales (from 2-20 km). This information was included in the conclusions (line 406): "In our work, at least in the sub-class meso-gama (2-20 km)."

In addition, we included some literature about the mesoscale range of NLLJs in the conclusions (lines 409-411): "These results agree with the classical theory on inertial oscillation linking NLLJ development to stable atmospheric conditions and horizontal pressure gradients in the mesoscale (e.g., Stensrud, 1996; Beyrich et al., 2006; Salio et al., 2007)."

3. Points to clarify:

Line 87-90: Does terrain discretization affects wind detection in reanalysis?

AC: Yes, the discretisation affects the representation of wind profiles and the NLLJ detection. We added it in the text (lines 87-89): "Reanalysis data can share similar biases for the near-surface wind profile and coarse vertical and spatial resolutions, including terrain discretization, are an additional contributing factor to those biases (Kalverla et al., 2019; Hallgren et al., 2020; Aird et al., 2021)."

Line 155: The vertical resolution apparently did not change (see Line 135). Was that a temporal smoothing?

AC: It is an adjustment of the vertical profile for comparability of the data from different instruments with different measurement heights and for reducing the

influence of turbulence. It now reads (line 158-162): "To that end, we interpolated all LIDAR vertically onto a single coarse-grained height profile as the middle between every measurement height from WL177 in Falkenberg. This way we obtain the same vertical resolution of 26.5 m for all LIDAR data. Second, we calculated hourly moving averages for all wind profiles. Both these approaches successfully decreased the number of false detections of NLLJs, e.g., those that are extremely short or false alarms arising from turbulent motion causing maxima in the vertical wind profiles."

Line 158: Can we affirm that nocturnal speed-ups in the wind with less than 1 hour are noise?

AC: The noise refers to turbulent motion on scales shorter than one hour. The word "noise" was changed to "turbulent motion" to avoid misunderstanding. The new sentence is (lines 160-162): "Both these approaches successfully decreased the number of false detections of NLLJs, e.g., those that are extremely short or false alarms arising from turbulent motion causing maxima in the vertical wind profiles."

Line 192: Limits for Ri depend on height evaluated?

AC: The limit does not depend on height. This information was included in the manuscript (line 195): "These theoretical limits are fixed and don't depend on height."

Line 256: How were the nights without NLLJ? Does the RI and DT develop differently from Fig.11a?

AC: The average dRi and dT have similar behavior when we include nights without NLLJs. This is a reflection of the higher stability of the atmosphere during night time. This information was included to the manuscript (267-268): "It is important to mention that the average dRi and dT had similar behavior when we included nights without NLLJs. This is a reflex of the higher stability of the atmosphere during night time".

Line 284: Is there any reference supporting the slackening geostrophic winds?
AC: We referred to the slackening of the geostrophic wind for one NLLJ event in the first version of the manuscript. Other long NLLJs have different changes in the geostrophic wind including both increasing and decreasing geostrophic winds (not shown). We made a small change to the manuscript to be specific (292-293): "The supergeostrophy, although already expected from theory, might at least in parts be explained by the slackening geostrophic winds in this example."

Technical corrections:
Line 41: Is "single" necessary?
AC: The word was removed.

Line 67: Revise sequence of citations;
AC: The citation order was corrected.

Line 81: Revise "With heights";

AC: The sentence was corrected: "Such measurements are usually also limited to the height of meteorological masts, typically up to 100m and sometimes up to 300 m, that are insufficient to fully characterize NLLJs."

Line 187: Standardize Ri formatting;

AC: The Ri formatting was corrected.

Line 228: Can not see "duration" in the Table 5;

AC: A change was made to the Table description and to the text (lines 234-236):

"Table 5 shows the number of nights with at least one NLLJ event, and the number of nights with NLLJs classified by their duration. The percentage in brackets is the frequency of occurrence of NLLJs relative to the total number of nights (first column) and relative to the total number of NLLJ nights (latter 4 columns)."

"Total number and percentage of nights with at least one NLLJ event and the number of nights with NLLJs classified by their duration. The percentage for the four latter columns are calculated from the number of NLLJ nights."

Line 275: Not clear what relative and absolute means

AC: We revised the text to (lines 283-285): "We identified a total shift in the direction of the mean absolute values of 34° for short events, 107° for long events and 132° for very long events, consistent with veering in an inertial oscillation.

Line 324-325: Revise units kW/h;

AC: Thanks, the units were corrected.

Line 329-333: Not clear paragraph. Differences are expected due to hub height and rated speed. Consider improving;

AC: We changed the paragraph for clarity (lines 340-344):

"Weak NLLJs (e.g. Figure 16d) were rare such that most NLLJs were strong enough to reach the cut-in wind speed of both wind turbines and allowed power generation. We observed NLLJ profiles with wind speeds bellow the cut-in threshold at the rotor heights of the E-126 (135 m) in 12 individual profiles, translating to 0.2% of all valid NLLJ profiles. For the V112 (94 m) this happened for 45 profiles or 0.7%. These differences between the two turbine types are primarily explained by the different rotor heights, since the cut-in wind speeds are identical."

Line 343-345: Not clear the shear/productivity gains. Consider improving;

AC: The values of the different layers were included for a better understanding of the paragraph (lines 355-359):

"The shear was substantially larger during NLLJs events. The mean wind shear in the mean rotor layer (50-150 m) had an increase of about 67% during NLLJ events. A similar value was found for V112 since that turbine type has a similar rotor layer (38-150 m). The higher E-126 (71.5-198.5 m) had a mean shear 53% lower, but the

increase during NLLJs was of 84%. We rarely saw negative shear values from cases like illustrated in Figure 16d and those that occurred were often below the cut-in wind speed, consistent with having relatively few and weak NLLJs at very low levels."

Reviewer 2

Primary comments:

* The authors compared wind data acquired by the LIDAR systems with wind observations from a sonic anemometer in Falkenberg. Figure 1b shows a high correlation between wind data from the two LIDARS in Falkenberg, but one overestimates, and the other underestimates the reference data from the anemometer. The authors pointed out the mast shadow caused such differences, but it holds high (around 2-3% of the average wind speed) after discarding wind speed data. I would like to see a more detailed discussion regarding such a result. The authors do not provide details on the local terrain (maybe it is complex). Are there other possible issues affecting the results, like the LIDAR operation mode or wind data amount?

AC: Figure 1a shows the comparison between the lidar WL177 and the anemometer. Lidar WL78 had similar behavior and is not shown. Figure 1b shows the comparisons between the three lidars at different heights, not against anemometer measurements. We improved the clarity of the caption of Figure 1 in response to the comment. The terrain of the measurement site in Falkenberg is flat and surrounded by agricultural land. Lindenberg is somewhat more complex with buildings and a small hill in the vicinity of the instrument. The shadow of the mast affecting the amount of valid data for validation and the LIDAR operation modes are discussed in detail by Steinheuer et al. (2022).

The following sentences were included in the manuscript:

Lines 111-113: "The terrain around the Falkenberg site is flat and surrounded by agricultural land. Lindenberg is located in a more complex area with some buildings and a small hill in the vicinity."

Lines 402-403: "The differences between both sites may depend on the different local characteristics (e.g., terrain) and, to a lesser extent, on the different operation modes of the LiDARs."

* I suggest the 500 m for maximum value in the x-axis scale of Figure 1b. A narrower scale lets the reader get more details from the plot for the altitudes in the range of wind turbines and easily understand the statement in lines 133-140.

AC: A new figure was made following the suggestions and will be included in the final manuscript.

* Why do the authors decide by hourly timeframes in lines 153-160 to filter the noisy behavior of wind speed data?

AC: After careful inspection of the profiles and tests for the detection of NLLJs, we decided to use hourly averaging that sufficiently smoothes signatures of turbulent motion in the vertical profiles of the winds. It decreased the number of false detections of NLLJs. We also changed the word "noise" to "turbulent motion" for clarity. The new sentence is (lines 160-162): "Both these approaches successfully decreased the number of false detections of NLLJs, e.g., those that are extremely short or false alarms arising from turbulent motion causing maxima in the vertical wind profiles."

* How long are the very short NLLJ? Less than an hour? I suggest the authors be more specific.
AC: Very short NLLJs are shorter than one hour (line 240). A change was made to the text to clarify the processing (lines 171-174), which were misleading before: "For an easier characterization of the NLLJs, we filled 10-minute gaps in between NLLJ detections i.e., we flagged non-NLLJ profiles in between two NLLJ profiles also as a NLLJ. This approach allowed us a better estimation of the duration of NLLJ events excluding very short perturbations. We than removed all NLLJ events shorter than 30 minutes to exclude short-lived events."

* Figure 11a shows the typical behavior of ndT and nRi during NLLJ event. How different is it from the regular condition (no-NLLJ)? Indeed, it would be interesting if non-NLLJ plots were also shown in Figure 11 with NLLJ plots.

AC: The general behavior of ndT and nRi is very similar also to no-NLLJ situations. This is a reflection of the larger stability of the atmosphere during nighttime. A figure with ndT and nRi including nights without NLLJs is attached to this reply. This information was included in the manuscript (lines 267-268): "It is important to mention that the average dRi and dT had similar behavior when we included nights without NLLJs. This is a reflex of the higher stability of the atmosphere during night time".

* Lines 323-328 - why compare absolute values and not normalized ones, as mentioned in the earlier paragraph?

AC: The normalized values were used for a better comparison between the histograms in Figure 15. The absolute values were later mentioned for a better understanding of the magnitudes.

* Line 328 - the authors say NLLJ events increase power production more strongly in taller wind turbines and high-rated power. I am not sure if it is entirely accurate because wind speed is the major but not the only factor to be considered.

AC: Indeed, other factors than wind speed affect the real power production. In this study we estimated the power production based on the power curves and the wind speed at the hub height, all else being equal. We revised the text to make this clearer (202-204): "It is known that factors other than the wind speed affect the power production. In this study, we estimated the stand-alone influence of wind speeds at hub height based on the power curves all else kept constant."

* I think the authors could merge the content presented in the Chapter Discussion and Conclusions

AC: Thank you for the suggestion. We prefer to keep the distinction between the two chapters for clarity.

Minor remarks:

- Line 25: "power consumption" or "installed power capacity"?

  (AC): The change was made to the text.

- I suggest a careful language revision to avoid minor issues like in Line 79 (…. measured at single stations …) where the word single is unnecessary and creates confusion;

  AC: „Single" was removed. Other minor issues were corrected, including the comments from both reviewers.

- I suggest the authors provide more details in Table 1 indicating the operation mode of each LIDAR instrument and the information regarding the instrumentation at the wind mast measurement station operating in Falkenberg.

  AC: The information was included to the table.

- Diurnal means "when the Sun is out" - opposite of nocturnal. I suggest the authors use "daily cycle" instead "diurnal cycle". For instance, the authors used "diurnal" in the caption of Figure 11.

  AC: „diurnal" was changed to „daily".

- line 344 - type instead typce

  AC: Corrected.